# Recursive self-embedded vocal motifs in wild orangutans

**Adriano R Lameira[1]\*, Madeleine E Hardus[2], Andrea Ravignani[3,4,5], Teresa Raimondi[6], Marco Gamba[6]**

[1]Department of Psychology, University of Warwick, Coventry, United Kingdom; [2]Independent Researcher, Warwick, United Kingdom; [3]Comparative Bioacoustics Group, Max Planck Institute for Psycholinguistics, Nijmegen, Netherlands; [4]Center for Music in the Brain, Department of Clinical Medicine, Aarhus University & The Royal Academy of Music Aarhus/Aalborg, Aarhus, Denmark; [5]Department of Human Neurosciences, Sapienza University of Rome, Rome, Italy; [6]Department of Life Sciences and Systems Biology, University of Turino, Torino, Italy

**Abstract** Recursive procedures that allow placing a vocal signal inside another of a similar kind provide a neuro-computational blueprint for syntax and phonology in spoken language and human song. There are, however, no known vocal sequences among nonhuman primates arranged in self-embedded patterns that evince vocal recursion or potential incipient or evolutionary transitional forms thereof, suggesting a neuro-cognitive transformation exclusive to humans. Here, we uncover that wild flanged male orangutan long calls feature rhythmically isochronous call sequences nested within isochronous call sequences, consistent with two hierarchical strata. Remarkably, three temporally and acoustically distinct call rhythms in the lower stratum were not related to the overarching rhythm at the higher stratum by any low multiples, which suggests that these recursive structures were neither the result of parallel non-hierarchical procedures nor anatomical artifacts of bodily constraints or resonances. Findings represent a case of temporally recursive hominid vocal combinatorics in the absence of syntax, semantics, phonology, or music. Second-order combinatorics, 'sequences within sequences', involving hierarchically organized and cyclically structured vocal sounds in ancient hominids may have preluded the evolution of recursion in modern language-able humans.

## eLife assessment

The paper represents a novel application of recursion theory to the long call vocalizations of orangutans to demonstrate repetitive, rhythmic sub-structuring. The authors use detailed acoustic analyses to show **compelling** evidence for self-embedded and nested isochronic motifs. These **fundamental** results have the potential to significantly advance current approaches used to compare nonhuman communication systems with human language.

## Introduction

Among the many definitions of recursion (**Martins, 2012**), the view that it represents the *repetition of an element or pattern within a self-similar element or pattern* has crossed centuries and disciplines, from **von Humboldt, 1836** and **Hockett, 1960** to **Mandelbrot, 1980** and **Chomsky, 2010**; from fractals in mathematics (**Mandelbrot, 1980**) to generative grammars in linguistics (**Chomsky, 2010**), from graphic (e.g., 'Print Gallery' by M. C. Escher) to popular art (e.g., 1940's Batman #8 comic book cover). Across varying terminologies, the common denominator across fields is that to re-curse (from the Latin

**\*For correspondence:**
adriano.lameira@warwick.ac.uk

**Competing interest:** The authors declare that no competing interests exist.

**eLife digest** Language is the most powerful communication tool known in nature. By combining a finite set of elements, it allows us to encode infinite messages. This enables communication about virtually anything, from alerting others to potential dangers, to recommending a favourite book. The prevailing theory of the last 70 years suggests that this ability rests on a computational process in the brain that is unique to humans, known as recursion.

Recursion enables humans to produce and place a language element or pattern of elements inside another element or pattern of the same kind. In this way, a clause can be embedded inside another 'carrier' clause to extend a thought, argument, or scenario, for example, "the dog, which chased the cat, was barking". While recursion offers a simple, yet potent, explanation for the endless possibilities of language, how and why recursion – and by extension language – emerged in humans but no other animals remains a mystery.

Lameira et al. observed vocal patterns in wild orangutans that appeared to be composed of different elements. As orangutans and other great apes are our closest living relatives, they represent the most realistic model for studying the ability of human ancestors to use and comprehend language. Therefore, Lameira et al. set out to determine if this was a case of vocal patterning embedded within a similar vocal pattern, which could indicate that recursion underpins production of these calls.

Analysing recordings of long calls made by wild male orangutans showed that they are organized as two layers, where calls with a regular beat (or tempo) are produced within another "carrier" call of a different tempo. Up to three different call types, each with their own signature tempo, can occur within the same carrier call. Further analysis confirmed these call types were unrelated to the carrier.

The findings of Lameira et al. demonstrate that orangutans produce recursive vocal sequences that could represent a possible precursor to recursion in humans, offering a potential avenue for studying how recursion, and ultimately language, evolved in humans. In the future, better understanding of how language evolved may help to refine machine learning algorithms that aim to recognize, predict or generate text.

to 're-run' or 're-invoke') is an operation that produces multiple, potentially infinite sets of items from one initial item or a finite set. This is achieved by nesting an item within itself or within another item of the same kind. Recursive patterns in everyday life are ubiquitous and include, for example, computer folders stored inside other computer folders, Russian dolls nested in each other, Romanesco broccoli's spirals arranged in a spiral, and the same number of minutes passed within the same number of hours (e.g., 12:12). Accordingly, recursion is not the simple repetition of a pattern or item on a single level (e.g., computer folders or Russian dolls side by side), but the placement of a pattern or item within itself (e.g., computer folders or Russian dolls inside each other), hence, generating different hierarchical levels or strata. This means that the same pattern or item is encountered at least at two different scales (e.g., 12 at the scale of hours, and 12 at the scale of minutes).

In language, although classically associated with syntax (*Chomsky, 2010*; *Idsardi et al., 2018*), recursion and its diagnostic self-embedded patterns have been recognized in phonology (*Bennett, 2018*; *Elfner, 2015*; *Barış and Revithiadou, 2009*; *Nasukawa, 2015*; *Nasukawa, 2020*; *Vogel, 2012*) and in verbal and non-verbal music (*Jackendoff, 2009*; *Koelsch et al., 2013*; *Martins et al., 2017*; *Sharma and Chimalakonda, 2018*), making these systems open-ended and theoretically inexhaustible. Recursive vocal sequences or structures in nonhuman primates could potentially inform incipient or transitional states of recursion along human evolution before the rise of modern language. However, their apparent absence, notably in great apes – our closest living relatives – has been interpreted as indicating that a neuro-cognitive or neuro-computational transformation occurred in our lineage but none other (*Hauser et al., 2002*). This absence of evidence has led some scholars to question altogether the role of natural selection for the emergence of language, tacitly favoring sudden 'hopeful monster' mutant scenarios (*Berwick and Chomsky, 2019*; *Bolhuis and Wynne, 2009*).

Decades-long debates on the evolution of language have carved around the successes and limitations of empirical comparative animal research (*Bolhuis et al., 2018*; *Bowling and Fitch, 2015*; *Corballis, 2014*; *Lameira, 2017a*; *Lameira and Call, 2020*; *Martins and Boeckx, 2019*; *Rawski et al., 2021*; *Townsend et al., 2018*). Syntax-like vocal combinatorics have been identified in some bird

(*Engesser et al., 2019*; *Engesser et al., 2016*; *Suzuki et al., 2016*; *Suzuki et al., 2017*) and primate species (*Jiang et al., 2018*; *Wang et al., 2015*; *Watson et al., 2020*), but vocal combinatorics were not claimed to be recursive nor was recursion directly tested. Three notable exceptions demonstrated recursion *learning* in nonhuman animal settings: (*Gentner et al., 2006*) in European starlings, *Ferrigno et al., 2020* in rhesus macaques, and *Liao et al., 2022* in crows. These studies show that animals can learn to recognize recursion in synthetic stimuli after dedicated human training in laboratory settings, but they do not show spontaneous production of recursive vocal combinatorics in naturalistic settings. Evidence of recursive vocal structures in wild animals (i.e., without human priming or intervention), notably in primates closely related to humans, such as great apes, would better inform what evolutionary precursors and processes could have led to the emergence of recursion in the human lineage.

## Direct structural approach to recursive combinatorics

A novel, direct approach to recursive vocal combinatorics in wild primates is desirable to help infer signal patterns that were recursive in some degree or kind in an extinct past, and molded subsequently into the recursive structures observed today in humans. By virtue of their own primitive nature, proto-recursive structures did not likely fall within modern-day classifications. Therefore, they will often fail to be predicted based on assumptions guided by modern language (*Kershenbaum et al., 2014*; *Miyagawa, 2021*). To this end, a structural approach is particularly advantageous based on the cross-disciplinary definition of recursion as 'the nesting of an element or pattern within a self-similar element or pattern'. First, no prior assumptions are required about species' cognitive capacities. High-level neuro-motor procedures are inferred only to the extent that these are directly reflected in how signal sequences are organized. For example, Chomsky's definition of recursion (*Chomsky, 2010*) can generate non-self-embedded signal structures, but these would be for that same reason operationally undetectable amongst other signal combinations. Second, no prior assumptions are required about signal meaning. There are no certain parallels between semantic content and word meaning in animals, but analyses of signal patterning allow us to identify the similarities between non-semantic (nonhuman) and semantic (human) combinatoric systems (*Lipkind et al., 2013*; *Sainburg et al., 2019*). The search for recursion can, hence, be made in the absence of lexical items, semantics, or syntax. Third, no prior assumptions are required about signal function. Under any evolutionary scenario, including punctuated hypotheses, ancestral signal function (whether cooperative, competitive, or otherwise) is expected to have derived or been leveraged by its proto-recursive structure. Otherwise, once present, recursion would not have been fixated among human ancestral populations. Accordingly, a structural approach opens the field to potentially untapped signal diversity in nature and yet unrecognized bona fide combinatoric possibilities within the human clade.

## Exploring recursive combinatorics in a wild great ape

Here, we undertake an explorative but direct structural approach to recursion. We provide evidence for recursive self-embedded vocal patterning in a (nonhuman) great ape, namely, in the long calls of flanged orangutan males in the wild. We conducted precise rhythm analyses (*De Gregorio et al., 2021*; *Roeske et al., 2020*) of 66 long call audio recordings produced by 10 orangutans (*Pongo pygmaeus wurmbii*) across approximately 2510 observation hours at Tuanan, Central Kalimantan, Indonesian Borneo. We identified five different element types that comprise the structural building blocks of long calls in the wild (*Hardus et al., 2009*; *Lameira and Wich, 2008*), of which the primary type are full pulses (*Figure 1A*). Full pulses do not, however, always exhibit uninterrupted vocal production throughout a long call (as during a long call's climax; *Spillmann et al., 2010*) but can break up into four different 'sub-pulse' element types: (i) grumble sub-pulses (quick succession of staccato calls that typically constitute the first build-up pulses of long calls; *Hardus et al., 2009*), (ii) sub-pulse transitory elements and (iii) pulse bodies (typically constituting pulses before and/or after climax pulses), and (iv) bubble sub-pulses (quick succession of staccato calls that typically constitute the last tail-off pulses of long calls) (*Figure 1A*). We characterized long calls' full- and sub-pulses' rhythmicity to determine whether orangutan long calls present a reiterated structure across different hierarchical strata. We extracted inter-onset intervals (IOIs; i.e., time difference between the start of a vocal element and the preceding one – $t_k$) from 8993 vocal long call elements (*Figure 1A*): 1930 full pulses (1916 after filtering for $0.025 < t_k < 5$ s), 757 grumble sub-pulses (731), 1068 sub-pulse transitory elements (374), 816 pulse bodies (11), and 4422 bubble sub-pulses (4193). From the extracted IOIs, we calculated

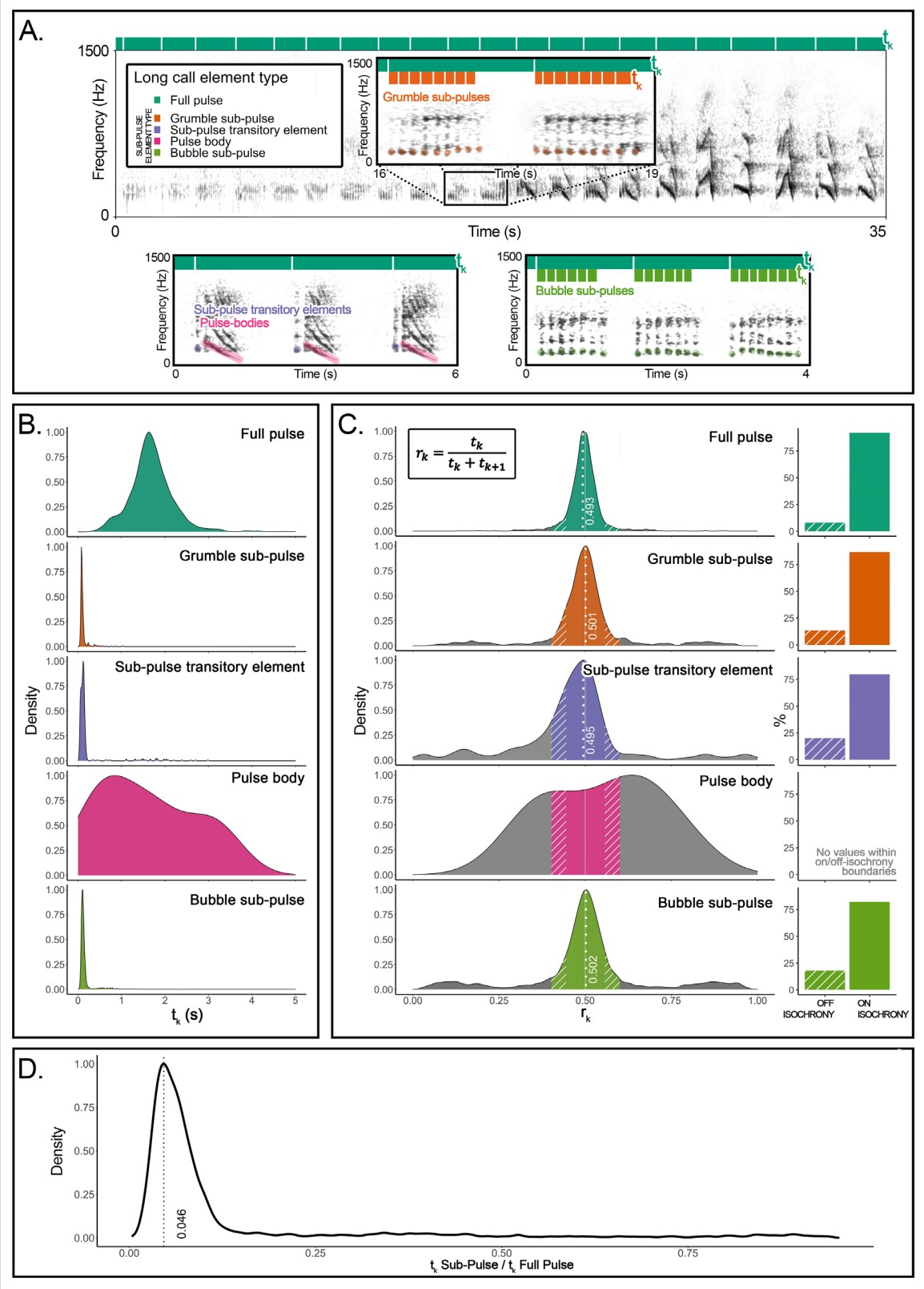

**Figure 1.** Organization and rhythmic features of orangutans' long calls. (**A**) Top: the spectrogram of a *full pulse* and its organization in sub-pulses (e.g., *grumble sub-pulses*). Below are the spectrograms of the three other sub-element types: *sub-pulse transitory* elements, *pulse bodies,* and *bubble sub-pulses*. Bars on the top of each spectrogram schematically quantify the durations of inter-onset intervals ($t_k$): dark green denotes the higher level of organization (*full pulse*). Orange (in the inset) and light green (bottom right) denote the lower-level organization (sub-pulse element types). (**B**)

*Figure 1 continued on next page*

*Figure 1 continued*

Probability density function showing the distributions of the inter-onset intervals ($t_k$) for each of the long call element types. (**C**) The distributions on the left show rhythm ratios ($r_k$) per element type as calculated on 12 flanged males for a total of 1915 full pulses and 5309 sub-pulses. Solid sections of the curves indicate on-isochrony $r_k$ values; striped sections indicate off-isochrony $r_k$ values. A solid white line indicates the 0.5 $r_k$ value corresponding to isochrony. White dotted lines denote the on-isochrony peak value extracted from the probability density function. Right: a bar plot per each element type shows the percentage of observations ($r_k$) falling into the on-isochrony boundaries (solid bars) or on off-isochrony boundaries (striped bars). The number of on-isochrony $r_k$ is significantly larger (GLMM, *full* vs *null*: Chisq = 2717.543, p<0.001) than the number of off-isochrony $r_k$ for all long call element types (*full pulse*: t-ratio = −25.164, p<0.001; *bubble sub-pulse*: t-ratio = −30.694, p<0.001; *grumble sub-pulse*: t-ratio = −14.526, p<0.001; *sub-pulse transitory element*: t-ratio = −3.148, p<0.001). *Pulse body* showed no $r_k$ values falling within the on-off-isochrony boundaries. (**D**) Distribution of a variable calculated as the ratio between the $t_k$ of a sub-pulse and the $t_k$ of the corresponding higher level of organization, the *full pulse*. We report the peak value of the curve (0.046) and tested the significance of the extent of the central quartiles, which was significantly smaller than peripheral quartiles (Wilcoxon signed-rank test: *W* = 2272, p<0.001).

their rhythmic ratio by dividing each IOI by its duration plus the duration of the following interval. We then computed the distribution of these ratios to ascertain whether the rhythm of long call full and sub-pulses presented natural categories, following published protocols (*De Gregorio et al., 2021*; *Roeske et al., 2020*; *Figure 1B–D*).

## Results

The density probability function of orangutan full pulses showed one peak ($r_k$ = 0.493) in close vicinity to a theoretically pure isochronic rhythm, that is, full pulses were regularly paced at a 1:1 ratio, following a constant tempo along the long call (*Figure 1C*). Our model (GLMM, *full* model vs *null* model: Chisq = 298.2876, df = 7, p<0.001; see *Supplementary file 1*) showed that pulse type, range of the curve (on-off-isochrony), and their interaction had a significant effect on the count of $r_k$ values. In particular, full pulses' isochronous peak tested significant (t.ratio = −15.957, p<0.0001), that is, the number of $r_k$ values falling inside the on-isochrony range was significantly higher than the number of $r_k$s falling inside the off-isochrony range (*Figure 1C*). Critically, three (of the four) orangutan sub-pulse element types – grumble sub-pulses, sub-pulse transitory elements, and bubble sub-pulses – also showed significant peaks (grumble sub-pulses: t.ratio = −5.940, p<0.0001; sub-pulse transitory elements: t.ratio = −4.048, p=0.0001; bubble sub-pulses: t.ratio = −10.640, p<0.0001) around pure isochrony (peak $r_k$: grumble sub-pulses = 0.501; sub-pulse transitory elements = 0.495; bubble sub-pulses = 0.502; *Figure 1C*). That is, sub-pulses were *regularly paced within regularly paced* full pulses, denoting isochrony within isochrony (*Figure 1C*) at different average tempi [mean $t_k$ (sd): full pulses = 1.696 (0.508); grumble sub-pulses = 0.118 (0.111); sub-pulse transitory elements = 0.239 (0.468); bubble sub-pulses = 0.186 (0.292); *Figure 1B*]. Overall, sub-pulses' $t_k$ was equivalent to 0.046 of their comprising full pulses (*Figure 1D*), which puts sub-pulses at an approximate ratio of 1:22 relative to

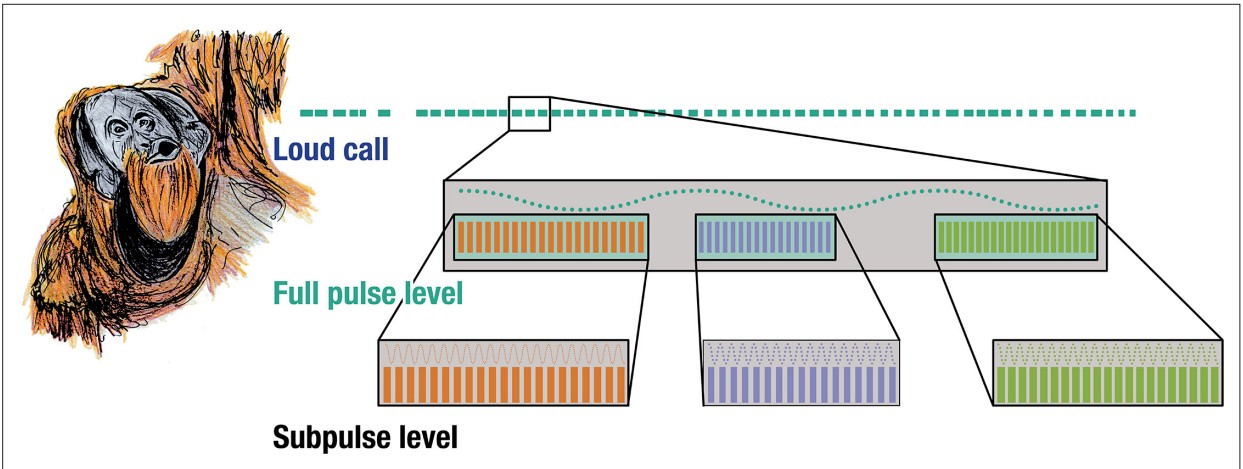

**Figure 2.** Isochrony nested within isochrony. Three acoustically distinct sub-pulse calls occurring at three distinct tempi nested within the same pulse-level tempo in wild flanged male orangutan long calls.

that of full pulses, the smallest categorical temporal rhythmic interval registered thus far in a vertebrate (*De Gregorio et al., 2021*; *Roeske et al., 2020*).

Permuted discriminant function analyses (*Mundry and Sommer, 2007*) (crossed, in order to control for individual variation) in R (*R Development Core Team, 2013*) based on seven acoustic measures extracted from grumble, transitory elements, and bubble sub-pulses confirmed that these represented indeed acoustically distinct sub-pulse categories, where the percentage of correctly classified selected cases (62.7%) was significantly higher (p=0.001) than expected (37%).

## Discussion

Rhythmic analyses of orangutan long calls reveal the presence of self-embedded isochrony in the vocal combinatorics of a wild great ape. Notably, we found that wild orangutan long calls exhibit two discernible structural strata – the full- and sub-pulse level – and three non-exclusive nested motifs in the form of [isochrony[A] [isochrony[a,b,c]]](*Figure 2*).

This is fundamentally distinct from a simple repetition of calls or call isochrony – when a call repeats linearly at a constant interval – which are common features in some animal sound communication systems (*De Gregorio et al., 2023*). Instead, we demonstrate how a vocal element repeated at a constant interval is itself composed by (one of three possible) vocal elements that also repeat themselves at a constant interval of different tempi.

The orangutans' production of recursive vocal motifs in the wild, and therefore, *without training*, is especially compelling in the context of the lab-based work that shows that nonhuman animals can learn recursion with training (*Ferrigno et al., 2020*; *Gentner et al., 2006*; *Liao et al., 2022*). Some aspects of these vocal combinatoric structures could be potentially learned as well (*Lameira et al., 2022*; *Lameira et al., 2016*; *Lameira et al., 2015*; *Lameira and Shumaker, 2019b*; *Wich et al., 2012*), but this study is agnostic on this matter because its design does not allow to single out learning effects. Nonetheless, results show that temporal recursion occurs spontaneously in the wild in great ape vocal communication.

### Can great apes hear recursive isochrony?

The observation that the long calls of wild orangutans possess isochronous characteristics raises questions about the ability of apes to perceive these signals. Humans perceive an acoustic pulse as a continuous pitch, instead of a rhythm, at rates higher than 30 Hz (i.e., 30 beats per second). Human and nonhuman great apes have similar auditory capacities (*Quam et al., 2015*), and there are limited skeletal differences in inner ear anatomy to suggest significantly distinct sensitivity, resolution, or activation thresholds in the time domain (*Quam et al., 2015*; *Spoor and Zonneveld, 1998*; *Stoessel et al., 2023*). Long call sub-pulses exhibited average rhythms at ~9.263 (sd: 3.994) Hz [i.e., $t_k$ = 0.184 (0.303) s]. Therefore, ear anatomy offers confidence that orangutans (and other great apes), like humans, perceive sub-pulse rhythmic motifs at these rates as such, that is, a train of signals, instead of one uninterrupted signal. Assuming otherwise would imply that auditory time resolution differs by more than one order of magnitude between humans and other great apes in the absence of obvious anatomical culprits.

### Can physiology fully explain recursive isochrony?

The occurrence of three non-exclusive recursive patterns (i.e., three acoustically distinct sub-pulse calls occurring at three distinct tempi nested within the same pulse-level tempo) substantially decreases the probability that recursion was the by-product of anatomic constraints, such as vocal fold oscillation, breath length, heartbeat, and other physiological processes or movements (*Pouw et al., 2020*). Such processes can generate frequency patterns nested within others; however, in these cases, sub-frequencies occur in the form of harmonics related to the reference (dominant) frequency and to each other by small whole-numbered multiples. Yet, the three observed rhythmic arrangements at the sub-pulse level were not related to the pulse level by any small integer ratios (i.e., 1/22). Also, some of these processes (e.g., vocal fold action) are oscillatory in nature, involving nested frequency waves. They are not combinatorial, involving nested sequences of events, as we report here.

Our data stimulate new questions about the relationship between oscillators and combinatoriality, which is difficult to investigate from an observational point of view in the wild. Hopefully, our results

will inspire new studies using controlled experimental settings to assess how oscillators and combinatoriality may be associated in ways potentially richer than thus far suspected. Together, our findings suggest that recursive isochrony is not the absolute result of raw mechanics but is instead likely generated or tampered with by, at least, one temporally recursive neuro-motor procedure.

## Can a linear algorithm produce recursive isochrony?

The occurrence of three non-exclusive recursive patterns drives down the likelihood that orangutans concatenate long call pulses and sub-pulses in a linear fashion and without bringing into play a recursive neuro-motoric process. To generate the observed vocal motifs linearly, three independent neuro-computational procedures would need to run in parallel. These three independent procedures would need to be indistinguishable, transposable, and/or interchangeable at the pulse level, whilst generating distinct rhythms and acoustics at the sub-pulse level. If theoretically possible at all, one would predict some degree of interference between the three linear procedures at the pulse level, manifested in some form of deviation around the isochrony peak. However, this was not observed; distribution of data points on and off isochrony was equivalent between pulses and sub-pulses.

## Precursor forms are not modern forms

Recursive self-embedded vocal motifs in orangutans indicate that vocal recursion among hominids is not exclusive to human vocal combinatorics, at least in the form of temporally embedded regular rhythms. This is not to suggest that orangutan recursive motifs exhibit *all* other properties that recursion exhibits in modern language-able humans, or that the two are the same, or equivalent. Further research will be necessary to fully unveil how orangutans use and control vocal recursion to form a clearer evolutionary picture. Expecting equivalence with language is, however, unwarranted as it would imply that no evolution would have occurred in over 10 million years since the split between the orangutan and human lineages. Any differences between our findings and recursion in today's syntax, phonology, or music do not logically reject the possibility that recursive isochrony represents an ancient, and perhaps ancestral, state for the evolution of vocal recursion within the hominid family.

## Implications for the evolution of recursion

Recursion and fractal phenomena are prevalent across the universe. Celestial and planetary movement, the splitting of tree branches, river deltas and arteries, and the morphology of bacteria colonies. Patterns within self-similar patterns are the norm, not the exception. This makes the seeming singularity of human recursion amongst animal vocal combinatorics all the more enigmatic. The discovery of recursive vocal patterns organized along two hierarchical temporal levels in a hominid besides humans suggests that 'sequences within sequences' may have been present in ancestral hominids, and hence, that second-order sequences may have predated the emergence of language in the human lineage.

Three major implications for the evolution of recursion in language apply. First, much ink has been laid on the topic. Yet, the possibility of self-embedded isochrony, or non-exclusive self-embedded patterns occurring within the same signal sequence, has on no account been formulated or conjectured as a possible state of recursive signaling, be it in vertebrates, mammals, primates, or otherwise, extant or extinct. This suggests that controversy may have been underscored by data-poor circumstances on vocal combinatorics in wild great apes, which only now start gathering comprehensive research effort (*Bortolato et al., 2023a*; *Bortolato et al., 2023b*; *Girard-Buttoz et al., 2022*; but see *Lameira et al., 2013a*). Resolution may come through a re-evaluation of previous studies with further related taxa and with experimental tests designed within a richer and more articulated panorama of observations on vocal combinatorics in wild great apes. Recursive vocal patterning in a wild great ape in the absence of syntax, semantics, phonology, or music opens a new charter for possible incipient and transitional states of recursion among hominids. The open discussion of what properties make a structure proto-recursive will be essential to move the state of knowledge past antithetical, dichotomous notions of how recursion and syntax evolved (*Berwick and Chomsky, 2019*; *Martins and Boeckx, 2019*).

Second, our findings invite renewed interest and reanalysis of primate vocal combinatorics in the wild (*Gabrić, 2022*; *Girard-Buttoz et al., 2022*; *Leroux et al., 2023*; *Leroux et al., 2021*). Given the dearth of such data, findings imply that it may be too hasty to discuss whether combinatorial capacities in primates or birds are equivalent to those engaged in syntax (*Engesser et al., 2015*; *Watson*

*et al., 2020*) or phonology (*Bowling and Fitch, 2015*; *Rawski et al., 2021*). Such classifications may be putting the proverbial cart before the horse; they are based on untested assumptions that may not have applied to proto-recursive ancestors (*Kershenbaum et al., 2014*; *Miyagawa, 2021*), for example, that syntax and phonology evolved as separate 'modules', that one attained modern form before the other, or that they evolved in hominids regardless of whether consonant-like and vowel-like calls were present or not.

Third, given that isochrony universally governs music and that recursion is a feature of music, findings could suggest a possible evolutionary link between great ape loud calls and vocal music. Loud calling is an archetypal trait in primates (*Wich and Nunn, 2002*) and among ancient hominids it could have preceded, and subsequently transmuted, into modern recursive vocal structures in humans, as found today in the form of song and chants. Given their conspicuousness, loud calls represent one of the most studied aspects of primate vocal behavior (*Wich and Nunn, 2002*), but their rhythmic patterns have only recently started to be characterized with precision (*Clink et al., 2020*; *De Gregorio et al., 2021*; *Gamba et al., 2016*). Besides our analyses, there are remarkably few confirmed cases of vocal isochrony in great apes (but see *Raimondi et al., 2023*). The behaviors that have been rhythmically measured with accuracy have been implicated in the evolution of percussion (*Fuhrmann et al., 2015*) and musical expression (*Dufour et al., 2015*; *Hattori and Tomonaga, 2020*), such as social entrainment in chimpanzees in connection with the origin of dance (*Lameira et al., 2019a*) (a capacity once also assumed to be neurologically impossible in great apes; *Fitch, 2017*; *Patel, 2014*). This opens the intriguing, tentative possibility that recursive vocal combinatorics were first and foremost a feature of proto-musical expression in human ancestors, later recruited and 're-engineered' for the generation of linguistic combinatorics. Future studies probing for recursive call sequence patterns in other orangutan vocal contexts and other great ape vocal behaviors could help test this idea.

## Concluding remarks

The presence of temporally recursive vocal motifs in a wild great ape revolutionizes how we can approach the evolution of recursion along the human lineage beyond all-or-nothing accounts. Future studies on primate vocal combinatorics, particularly undertaking a structural approach and in the wild, offer promising new paths to empirically assess possible precursors and proto-states for the evolution of recursion within the hominid family, also adding temporal recursion as a new layer of analysis. These crucial data on the evolution of recursion, language, and cognition along the human lineage will materialize if, as stewards of our planetary co-habitants, humankind secures the survival of nonhuman primates and the preservation of their habitats in the wild (*Estrada et al., 2022*; *Estrada et al., 2017*; *Laurance, 2013*; *Laurance et al., 2012*).

# Materials and methods
## Study site

We conducted our research at the Tuanan Research Station (2°09′S; 114°26′E), Central Kalimantan, Indonesia. Long calls were opportunistically recorded from identified flanged males (*P. pygmaeus wurmbii*) using a Marantz Analogue Recorder PMD222 in combination with a Sennheiser Microphone ME 64 or a Sony Digital Recorder TCD-D100 in combination with a Sony Microphone ECM-M907.

## Acoustic data extraction

Audio recordings were transferred to a computer with a sampling rate of 44.1 kHz. Seven acoustic measures were extracted directly from the spectrogram window (window type: Hann; 3 dB filter bandwidth: 124 Hz; grid frequency resolution: 2.69 Hz; grid time resolution: 256 samples) by manually drawing a selection encompassing the complete long call (sub)pulse from onset to offset using Raven interactive sound analysis software (version 1.5, Cornell Lab of Ornithology). These parameters were duration (s), peak frequency (Hz), peak time, peak frequency contour average slope (Hz), peak frequency contour maximum slope (Hz), average entropy (Hz), and signal-to-noise ratio (NIST quick method). Please see the software's documentation for a full description of the parameters. Acoustic data extraction complemented the classification of long calls elements, both at the pulse and sub-pulse levels, based on close visual and auditory inspection of spectrograms, both based on elements' distinctiveness between each other as well as in relation to the remaining cataloged orangutan call

repertoire (*Hardus et al., 2009*; see also *Supplementary files 2–5*). Of these parameters, duration and peak frequency in particular have been shown to be resilient across recording settings (*Lameira et al., 2013b*) and adequately represent variation in the time and frequency axes (*Lameira et al., 2017b*).

## Rhythm data analyses

IOIs (IOI's = $t_k$) were only calculated from the begin time (s) of each full- and sub-pulse long call elements using Raven interactive sound analysis software, as above explained. $t_k$ was calculated only from subsequent (full/sub) pulse elements of the same type. Ratio values ($r_k$) were calculated as $t_k/(t_k + t_{k+1})$. Following the methodology of *Roeske et al., 2020* and *De Gregorio et al., 2021*, to assess the significance of the peaks around isochrony (corresponding to the 0.5 $r_k$ value), we counted the number of $r_k$s falling inside the on-isochrony ranges (0.440 < $r_k$ < 0.555) and off-isochrony ranges (0.400 < $r_k$ < 0.440 and 0.555 < $r_k$ < 0.600), symmetrically falling at the right and left sides of 1:1 ratios (0.5 $r_k$ value). We tested the count of on-isochrony $r_k$s versus the count of off-isochrony $r_k$s, per pulse type, with a GLMM for negative-binomial family distributions, using *glmmTMB* R library. In particular, we built a *full* model with the count of $r_k$ values as the response variable, the pulse type in interaction with the range the observation fell in (on- or off- isochrony) as predictors. We added an offset weighting the $r_k$ count based on the width of the bin. The individual contribution was set as random factor. We built a *null* model comprising only the offset and the random intercepts. We checked the number of residuals of the full and *null* models, and compared the two models with a likelihood ratio test (ANOVA with 'Chisq' argument). We calculated p-values for each predictor using the R *summary* function and performed pairwise comparisons for each level of the explanatory variables with *emmeans* R package, adjusting all p-values with Bonferroni correction. We checked normality, homogeneity (via function provided by R. Mundry), and number of the residuals. We checked for overdispersion with *performance* R package (*Lüdecke et al., 2021*). Graphic visualization was prepared using R (*R Development Core Team, 2013*) packages *ggplot2* (*Wickham, 2009*) and *ggridges* (*Wilke, 2022*). Data reshape and organization were managed with *dplyr* and *tidyr* R packages.

## Acoustic data analyses

Permutated discriminant function analysis with cross-classification was performed using R and a function provided by R. Mundry (*Mundry and Sommer, 2007*). The script was pdfa.res=pDFA.crossed ( test.fac="Sub-pulse-type", contr.fac="Individual.ID", variables=c("Delta.Time", "Peak.Freq", "Peak. Time", "PFC.Avg.Slope", "PFC.Max.Slope", "Avg.Entropy", "SNR.NIST.Quick"), n.to.sel=NULL, n.sel=100, n.perm=1000, pdfa.data=xdata). These analyses assured that long call elements, at the pulse and sub-pulse levels, indeed represented biologically distinct categories.

## Acknowledgements

We thank the Indonesian Ministry of Research and Technology, the Indonesian Ministry of Environment and Forestry, the Indonesian Ministry of Home Affairs, the Directorate General of Natural Resources and Ecosystem Conservation, and the former Directorate General of Forest Protection and Nature Conservation for authorization to carry out research in Indonesia; the Universitas National for supporting the project and acting as sponsors and counter-partners; the Bornean Orangutan Survival Foundation and the MAWAS Programme in Palangkaraya for their support and permission to stay and work in the MAWAS Reserve. ARL was supported by the UK Research & Innovation, Future Leaders Fellowship grant agreement number MR/T04229X/1.

## Additional information

### Funding

| Funder | Grant reference number | Author |
|---|---|---|
| UK Research and Innovation | MR/T04229X/1 | Adriano R Lameira |

| Funder | Grant reference number | Author |
|--------|------------------------|--------|

The funders had no role in study design, data collection and interpretation, or the decision to submit the work for publication.

## Author contributions

Adriano R Lameira, Conceptualization, Resources, Data curation, Formal analysis, Supervision, Funding acquisition, Validation, Investigation, Visualization, Methodology, Writing - original draft, Project administration, Writing – review and editing; Madeleine E Hardus, Resources, Data curation, Investigation, Project administration, Writing – review and editing; Andrea Ravignani, Resources, Methodology, Writing – review and editing; Teresa Raimondi, Resources, Data curation, Formal analysis, Investigation, Visualization, Methodology, Writing – review and editing; Marco Gamba, Conceptualization, Resources, Formal analysis, Supervision, Validation, Investigation, Visualization, Methodology, Writing - original draft, Project administration, Writing – review and editing

## Author ORCIDs

Adriano R Lameira 
Marco Gamba 

## Ethics

This study was performed in strict accordance with the Indonesia law and the recommendations of the Indonesian Institute of Sciences (LIPI), the Univesitas National (UNAS) and the Borneo Orangutan Survival Foundation (BOS).

Reviewer #1 (Public Review): https://doi.org/10.7554/eLife.88348.3.sa1
Reviewer #3 (Public Review): https://doi.org/10.7554/eLife.88348.3.sa2
Author Response https://doi.org/10.7554/eLife.88348.3.sa3

# Additional files

## Supplementary files

- Supplementary file 1. GLMM statistical details.
- Supplementary file 2. Example set of male orangutan grumbles.
- Supplementary file 3. Example set of male orangutan transitory elements and pulse body.
- Supplementary file 4. Example set of male orangutan full pulses.
- Supplementary file 5. Example set of male orangutan bubbles.
- MDAR checklist

## Data availability

Raw data available at: https://osf.io/w3ne5/.

The following dataset was generated:

| Author(s) | Year | Dataset title | Dataset URL | Database and Identifier |
|-----------|------|---------------|-------------|-------------------------|
| Lameira A | 2023 | Recursive self-embedded vocal motifs in wild orangutans | https://osf.io/w3ne5/ | Open Science Framework, w3ne5 |

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
