## [Editor Report · eLife assessment]

The paper represents a novel application of recursion theory to the long call vocalizations of orangutans to demonstrate repetitive, rhythmic sub-structuring. The authors use detailed acoustic analyses to show **compelling** evidence for self-embedded and nested isochronic motifs. These **fundamental** results have the potential to significantly advance current approaches used to compare nonhuman communication systems with human language.

---

## [Referee Report · Reviewer #1 (Public Review)]

This study investigates the structuring of long calls in orangutans. The authors demonstrate long calls are structured around full pulses, repeated following a regular tempo (isochronic rhythm). These full pulses are themselves structured around different sub-pulses, themselves repeated following an isochronic rhythm. The authors argue this patterning is evidence for self-embedded, recursive structuring in orangutang long calls.

The analyses conducted are robust and compelling and they support the rhythmicity the authors argue is present in the long calls. Furthermore, the authors went above and beyond and confirmed acoustically the sub-categories identified were accurate.

---

## [Referee Report · Reviewer #3 (Public Review)]

Summary: This paper presents evidence of recursive self-embedding in the vocalization structure of orangutans, using fine-grained acoustical analysis. It proves the existence of isochrony nested in isochrony in the motifs produced by a nonhuman vocal system.

Strengths: Very clear written, clear analysis, excellent responses to the Reviewers.

Weaknesses: Jargonous language may be reduced. A video showing the sound as it unfolds and the spectrogram (as in Fig 1A) of the long call could be useful to best exemplify the results.

---

## [Author Response]

The following is the authors’ response to the original reviews.

Thank you very much for forwarding these two important reviews on our paper. Please find hereby our point-by-point responses addressing the ideas, arguments and points of concern raised by the reviewers. We provide explanation of how these points have been incorporated in the paper.

We feel the review process has been a useful exercise and that the paper has greatly benefited in terms of clarity and accessibility. It is our hope that our findings may ignite renewed interest on unexplored and “unexpected” aspects of great ape vocal communication, inspire novel research, and invite bold new advances on the long-standing puzzle of language origins and evolution.In several relevant sections, we have also sought to explicitly address the point of doubt raised in eLife’s editorial assessment, published alongside the reviewed preprint of our paper. The editorial assessment stated that “…However the evidence provided to support the major claims of the paper is currently incomplete. Specifically, it is not yet clear how the rhythmic structuring found in these long calls is more similar to human language recursion per se rather than isochrony as a broader, more common phenomenon.” To directly clarify this point, we provide now various examples of how recursion is distinct from repetition, using everyday objects for an intuitive understanding (e.g., lines 43-51). We have also expanded the discussion to better contextualise and clarify the implications of our findings on language evolution theory. We hope this will help addressing the implicit request for clarification in the previous editorial assessment.

Thank you very much for your kind and dedicated attention in the processing of our study.

**Public Reviews:**

**Reviewer #1 (Public Review):**
This study investigates the structuring of long calls in orangutans. The authors demonstrate long calls are structured around full pulses, repeated following a regular tempo (isochronic rhythm). These full pulses are themselves structured around different sub-pulses, themselves repeated following an isochronic rhythm. The authors argue this patterning is evidence for self-embedded, recursive structuring in orangutang long calls.The analyses conducted are robust and compelling and they support the rhythmicity the authors argue is present in the long calls. Furthermore, the authors went above and beyond and confirmed acoustically the sub-categories identified were accurate.

We thank the reviewer for this important support regarding our methods and findings.

However, I believe the manuscript would benefit from a formal analysis of the specific recursive patterning occurring in the long call. Indeed, as of now, it is difficult for the reader to identify what the authors argue to be recursion and distinguish it from simple repetitions of motifs, which is essential.

We agree with the reviewer that the distinction between repetition and recursion is very important for the adequate interpretation of our findings. Following the reviewer’s point (and the Editorial Assessement), we have now rephrased several passages in the initial paragraph of the paper for added clarity, where recursion is introduced and explained. We now also provide various new examples of recursion in everyday life and popular culture to better illustrate in an easy and accessible way the fundamental nature of recursion. We then use two of these common examples (computer folders and Russian dolls) to specifically distinguish repetition from recursion.

Although the authors already discuss briefly why linear patterning is unlikely, the reader would benefit from expanding on this discussion section and clarifying the argument here (a lay terminology might help).

Corrected accordingly.

I believe an illustration here might help. In the same logic, I believe a tree similar to the trees used in linguistics to illustrate hierarchical structuring would help the reader understand the recursive patterning in place here. This would also help get the "big picture", as Fig 1A is depicting a frustratingly small portion of the long call.

We completely understand the reviewer’s concern here. As proposed by the reviewer, and in addition to changes in the Introduction (see above) and Discussion (see below), we have now added a new figure in the Discussion to help the reader get the “big picture” of our findings.

We have also made revisions throughout the Introduction and Discussion to simplify the text, clarify our exposition and facilitate the reader better and intuitively understand the nature and relevance of our results.

Notwithstanding these comments, this paper would provide crucial evidence for recursion in the vocal *production* of a non-human ape species. The implication it would have would represent a key shift in the field of language evolution. The study is very elegant and well-constructed. The paper is extremely well written, and the point of view adopted is original, well-argued and compelling.

We are humbled by the reviewer’s words, and we thank the reviewer for attributing these qualities to our paper. This feedback reassures us of the disruptive potential that these and similar future findings may have on our understanding of language evolution.

**Reviewer #2 (Public Review):**
I am not qualified to judge the narrow claim that certain units of the long calls are isochronous at various levels of the pulse hierarchy. I will assume that the modelling was done properly. I can however say that the broad claims that (i) this constitutes evidence for recursion in non-human primates, (ii) this sheds light on the evolution of recursion and/or language in humans are, when not made trivially true by a semantic shift, unsupported by the narrow claims. In addition, this paper contains errors in the interpretation of previous literature.

We report the first confirmed case of “vocal sequences within vocal sequences” in a wild nonhuman primate, namely a great ape. The currently prevailing models of language evolution often rest on the (purely theorical) premise that such structures do not exist in any animal bar humans. We find the discovery of such structures in a wild great ape exciting, remarkable, and promising. We regret that the reviewer does not share this sentiment with us. We feel that the statement that these findings are trivial and narrow is unfounded.

In order to clarify and better communicate the significance of our findings, we now explain in more detail in the Introduction and Discussion how the discovery of nested isochrony in wild orangutans promises to stimulate new series of studies in nature and captivity. Our findings dovetail nicely with previous captive studies that have shown that animals can learn how to recognise recursive patterns and invite new research efforts for the investigation of recursive abilities in the wild and in the absence of human priming and in nonhuman primates.

The main difficulty when making claims about recursion is to understand precisely what is meant by "recursion" (arguably a broader problem with the literature that the authors engage with). The authors offer some characterization of the concept which is vague enough that it can include anything from "celestial and planetary movement to the splitting of tree branches and river deltas, and the morphology of bacteria colonies". With this appropriately broad understanding, the authors are able to show "recursion" in orangutans' long calls. But they are, in fact, able to find it everywhere.

The reviewer is correct in highlighting that recursion is ubiquitous in nature and this is something that we explicitly state in the paper. This only makes it the more surprising that, when it comes to vocal combinatorics, recursion has only been described in human language and music, but in no other animals. If studies providing such evidence are known to reviewer, we kindly request their corresponding references.

In the new revised version, we have paid attention to this aspect raised by the reviewer, and we have sought to disambiguate that our observations pertain to temporal recursion. This clarification will hopefully allow a better understanding of our results.

The sound of a plucked guitar string, which is a sum of self-similar periodic patterns, count as recursive under their definition as well.

The example pointed out here by reviewer is factually correct; sound harmonics represent a recursive pattern of a fundamental frequency. (In fact, we explain this phenomenon in the Discussion.) The reviewer’s comment seems to offer an analogy to oscillatory phenomena in the physiology of the vocal folds, and so, it is misplaced with regards to our present study, which focused vocal sequences. Admittedly, this misinterpretation may have been implicitly caused by our wording and we apologise for this. We now refer to “vocal combinatorics” instead of “vocal production” throughout the paper to avoid the reader considering that our findings pertain to the physiology of the vocal folds.

One can only pick one's definition of recursion, within the context of the question of interest: evolution of language in humans. One must try to name a property which is somewhat specific to human language, and not a ubiquitous feature of the universe we live in, like self-similarity. Only after having carved out a sufficiently distinctive feature of human language, can we start the work of trying to find it in a related species and tracing its evolutionary history. When linguists speak of recursion, they speak of in principle unbounded nested structure (as in e.g., "the doctor's mother's mother's mother's mother ..."). The author seems to acknowledge this in the first line of the introduction: "the capacity to *iterate* a signal within a self-similar signal" (emphasis added). In formal language theory, which provides a formal and precise definition of one notion of recursivity appropriate for human language, unbounded iteration makes a critical difference: bounded "nested structures" are regular (can be parsed and generated using finite-state machines), unbounded ones are (often) context-free (require more sophisticated automaton). The hierarchy of pulses and sub-pulses only has a fixed amount of layers, moreover the same in all productions; it does not "iterate".

The reviewer explains here how recursion, in its fully fledged form in modern language(s), is defined by linguistics. We fully agree and do not contest such descriptions and definitions in any way. These descriptions and definitions aim to describe how recursion operates today, not how it evolved. Nor do these descriptions and definitions generate data-driven, testable predictions about precursors or proto-states of recursion as used by modern language-able humans. This is scientifically problematic and heuristically unsatisfying regarding the open question of language evolution.

Following human-specific definitions for recursion, as proposed by the reviewer, cannot per se be used to undertake a comparative approach to evolution because they leave nothing to compare recursion with in other (wild) species. Using human-specific definitions unavoidably leads to black-and-white notions that language is always absolutely present in humans and always absolutely absent in other animals, regardless of their degree of relatedness to humans. It is unpreventable that these descriptions flout foundational principles of evolution, such as descent with modification and shared ancestry.

This conceptual problem is not new. Less than a century ago, it was believed that humans were the only tool-user (thousands of examples are known today in nonhuman animals, including fish and invertebrates), and later, that humans were the only cultural animal (today it is known that migrating caribou and fruit flies can establish traditions based on social learning). We must follow in the footsteps of those who have helped redefine human nature in the past. As famously stated by Louis Leakey when presented with evidence for chimpanzee tool-use collected by Jane Goodall, “Now we must redefine tool, redefine man, or accept chimpanzees as human”. Therefore, as a matter of course, we must redefine recursion, embracing empirically (other than purely theoretically) definitions that allow recursion to take on forms and functions different from that of modern language-able humans.

Another point is that the authors don't show that the constraints that govern the shape of orangutans long calls are due to cognitive processes.

The reviewer is indeed correct. This does not, however, refute our findings. We do not directly show that cognitive processes govern recursion in orangutan long calls. Instead, we show that the observed patterns cannot be explained by simple bodily or motoric processes, excluding therefore low-level explanations. With more than 50 years of accumulated field experience in primatology, this was the only possible way that our team found to go about conducting research and analyses on natural behaviour, in the wild, with a critically endangered primate. We would be very interested in learning from the reviewer what ethical and non-invasive methods, specific locations in the wild, and type of behavioural or socio-ecological data could be otherwise viably used to demonstrate what the reviewer requests. If other scientists believe that the patterns observed in wild orangutan long calls – three independent, but simultaneously-occurring recursive motifs – can be generated based on low-level physiological mechanisms alone, the burden of proof resides with them.

Any oscillating system will, by definition, exhibit isochrony.

We disagree with this statement. The example provided above by the reviewer him/her-self disproves the statement: a guitar string when struck is an oscillating system but it is not isochronic nor is it combinatorial. Isochrony cannot be established with single events, only with event sequences (in practice, ideally >3).

For instance, human trills produce isochronouns or near isochronous pulses. No cognitive process is needed to explain this; this is merely the physics of the articulators. Do we know that the rhythm of the pulses and sub-pulses in orangutans is dictated by cognition as opposed to the physics of the articulators?

The reviewer seems to misinterpret our results here. Our focus is on vocal combinatorics, not vocal fold oscillation (see previous response). We have now reworded all instances where the text could be unclear.

Even granting the authors' unjustified conclusion that wild orangutans have "recursive" structures and that these are the result of cognition, the conclusions drawn by the authors are too often fantastic leaps of induction. Here is a cherry-picked list of some of the far-fetched conclusions:

"our findings indicate that ancient vocal patterns organized across nested structural strata were likely present in ancestral hominids". Does finding "vocal patterns organized across nested structural strata" in wild orangutans suggest that the same were present in ancestral hominids?

Following the reviewer’s comment, we have now rephrased and toned down this passage, stating that such structures “may have been present” in ancestral hominids. We are grateful to the reviewer for this comment.

"given that isochrony universally governs music and that recursion is a feature of music, findings (sic.) suggest a possible evolutionary link between great ape loud calls and vocal music". Isochrony is also a feature of the noise produced by cicadas. Does this suggest an evolutionary link between vocal music and the noise of cicadas?

We apologise, but it is unclear what the reviewer is exactly suggesting or proposing here. It seems as though it is believed that cicadas are as phylogenetically related to humans as great apes are. Our last common ancestor with great apes diverged about 10mya, but with cicadas 600mya. The last common ancestor with great apes was a great ape (or hominid). The human-cicada last common ancestor would have looked like a worm (it is probable it would already have a nervous bulge at the head, or “brain”). In order to avoid similar misinterpretations, we have now clarified in several instances that our study and interpretation of results are based on shared ancestry within the Hominid family.

It seems that the reviewer may be also misinterpreting our findings. We do not simply report isochrony in a wild great ape (multiple references for isochronous calls in primate are provided in the Discussion). We report isochrony within isochrony in three non-exclusive rhythmic arrangements. In case the reviewer knows of a study on cicadas, or any non-human species, showing recursive sound combinatorics of this nature, we kindly request the citation. We can only hope that such new cases may be gradually unveiled in wild animals to help propel our general understanding of possible ways of how insipient recursive vocal combinatorics in ancient hominids could have given rise to recursion as used today by language-able modern humans.

Finally, some passages also reveal quite glaring misunderstandings of the cited literature. For instance:"Therefore, the search for recursion can be made in the absence of meaning-base operations, such as Merge, and more generally, semantics and syntax". It is precisely Chomsky's (disputable) opinion that the main operation that govern syntax, Merge, has nothing to do with semantics. The latter is dealt within a putative conceptual-intentional performance system (in Chomsky's terminology), which is governed by different operations.

Following the reviewer’s comment, we have now removed “meaning-base operations, such as Merge, and more generally” from the target sentence in order to avoid confusion. Thank you.

"Namely, experimental stimuli have consisted of artificial recursive signal sequences organized along a single temporal scale (though not structurally linear), similarly with how Merge and syntax operate". The minimalist view advocated by Chomsky assumes that mapping a hierarchichal structure to a linear order (a process called linearizarion) is part of the articulatory-perceptual system. This system is likewise not governed by Merge and is not part of "syntax" as conceived by the Chomskyan minimalists.

Following the reviewer’s comment, we have not omitted the target sentence for added clarity.

**Reviewer #1 (Recommendations For The Authors):**
L55-67: I feel there is a step missing in the logic of the argumentation here.The studies cited by the authors here are mostly about syntactic-like structuring but not recursion. Hence when the authors mention in the next sentence that these studies investigate the perception of recursive signalling, it seems incorrect. I agree with the logic, but the references do not seem appropriate. I would further suggest that if there are no other references, that would make the introduction of the study here even easier: there is very little work investigating this capacity in non-human animals, let alone on a production perspective, therefore, the study conducted here is paramount and fills this important gap in the literature.

We are grateful to Reviewer #1 for these comments, and we are honoured to hear that our findings are filling a literature gap. We have now carefully revised the manuscript, hopefully, streamlining our line of reasoning and improving the paper’s overall readability. We agree that there is very little work investigating the spontaneous “production” of recursion in nonhuman animals. We decided to better detail the logic of our paper by clarifying the difference between recursion and repetition and clarifying that the motifs that we identify in wild orangutan represent a case of "temporal recursion".

L59: Johan J should be removed (same in discussion).

Removed, thanks.

L60: For example is repeated twice, here and L55.

We have rephrased this part of the manuscript, thanks.

L72-73: If we consider the Watson et al., 2020 study an example of recursive perception (which I do not think is true), this was conducted using a passive design - i.e. with no active training.

We have rephrased this part of the manuscript, thanks.

L240-241: Again, non-adjacent dependency processing does not equal recursion.

We agree that non-adjacent dependency processing does not equal recursion. We have now clarified this section accordingly.

L269: one *of* the most.

Corrected, thanks.

L296: add space after settings.

Corrected, thanks.

**Reviewer #2 (Recommendations For The Authors):**
In addition to the public portion of the review, I advise the authors' to substantially alter their style of writing. The language used is not accurate and the intended meaning is often not clear. This makes it hard for any reader to follow the authors' reasoning fully. Below I list only a few of the egregious examples but the examples abound:"this hints at a neuro-cognitive or neuro-computational transformation in the human brain" what meaning do the author assign to "neuro-cognitive" and "neuro-computational" ? what difference do they place between the two (so that they would be disjoined.) ? What "transformation" are we talking about ? From what to what ?" However, recursive signal structures can also unfold in other manners, such as across nested temporal scales and in the absence of semantics (Fitch, 2017a), as in music." what is meant here by nested temporal scales ?"The simultaneous occurrence of non-exclusive recursive patterns excludes the likelihood that orangutans concatenate long calls and their subunits in linear structure without any recursive processes": isn't there a more straightforward way to say "excludes the likelihood"? What is meant by "non-exclusive recursive patterns"?

It seems that Reviewer #2 does not share our writing style. Nonetheless, we have tried to meet the reviewer halfway, clarifying throughout the new revised version our definitions, our line of argument, our motivations, our results, the context of our findings in what is known about recursion in animals, and the implication of our discovery for language evolution theory.